# Modification of the Biorelevant Release Testing of Esophageal Applied Mucoadhesive Films and Development of Formulation Strategies to Increase the Mucosal Contact Time

**DOI:** 10.3390/pharmaceutics16081021

**Published:** 2024-08-01

**Authors:** Friederike Brokmann, Paul Simonek, Christoph Rosenbaum

**Affiliations:** Department of Biopharmaceutics and Pharmaceutical Technology, Institute of Pharmacy, University of Greifswald, Felix-Hausdorff-Straße 3, 17489 Greifswald, Germany

**Keywords:** esophagus, local drug targeting, biorelevant in vitro model, mucoadhesive polymer, films, peristalsis, saliva flow, esophageal transport

## Abstract

The increasing prevalence of esophageal disease highlights the clinical relevance of novel, long-lasting mucoadhesive oral dosage forms. The EsoCap device enables targeted local application of films in the esophagus. Biorelevant test systems such as EsoPeriDiss are essential for early formulation development. To this end, the developed and already described release model for simulating the esophagus is being further developed for its potential for biorelevant mapping of the application site through complete tempering and investigation of biorelevant release media. Particularly viscous saliva formulations led to an extension of the retention time. In addition, possible formulation strategies for increasing the retention time of esophageal applied films are being evaluated, such as different film thicknesses, polymer grades and the influence of different active ingredient properties on the retention time. For highly soluble active ingredients, the film thickness represents an option for extending the retention time, while for less soluble substances, the choice of polymer grade may be of particular interest.

## 1. Introduction

The number and prevalence of esophageal diseases such as gastroesophageal reflux disease (GERD), eosinophilic esophagitis (EoE), Barrett’s esophagus and esophageal cancer have increased significantly in recent years [1,2,3]. To meet the challenges of therapy, such as in the case of EoE, the long-lasting local application of active substances at the site of the disease is a key factor for success [4,5]. The particularly short transit times of oral dosage forms, which range from 7 to 14 s depending on type, viscosity, amount and body position, pose a major challenge [6,7]. Novel delivery systems, such as the EsoCap concept, which has already been successfully tested in a phase II clinical trial, are intended to enable prolonged drug delivery in the esophagus through the local application of a mucoadhesive film that forms a viscous gel upon swelling [8,9].

To assess the biorelevant release behaviour of newly developed films for esophageal application, a biorelevant dissolution model was developed with the so-called EsoPeriDiss, which can simulate the flow rate, inclination and peristalsis of the esophagus [10,11]. The anatomy and physiology of the esophagus, an approximately 25 cm long collapsed muscular tube, was simulated for this purpose [12]. A healthy adult swallows between 100 and 600 times a day, with one-third of swallows being due to the ingestion of food or drink, which triggers a primary peristaltic wave [13,14]. Less than 10% of swallows occur at night [15]. The peristaltic wave propagates as a contractile wave with a velocity of 2–6 cm/s in the distal esophagus. In addition, involuntary secondary peristalsis may be triggered, e.g., as a self-cleaning reflex of the esophagus [16]. A large part of the daily swallowing act is accounted for by basal salivary secretion, which ranges from 500–2500 mL per day and is mainly due to the three paired salivary glands. In addition, there are other small salivary glands in the submucosal layer of the esophagus which are more prominent in the upper and lower parts of the esophagus and produce saliva containing bicarbonate, e.g., for acid resistance [17]. The small glands and the sublingual gland are purely mucous glands that produce a mucin-rich secretion. Together they are responsible for 80% of the daily mucin production. In addition, saliva is 97–99% water, in which electrolytes, lipids and glycoproteins are dissolved [15,18]. Ionic components include phosphate and bicarbonate, which contribute to the buffering capacity of saliva, as well as sodium, potassium, magnesium, zinc, calcium, chloride, fluoride, iodide and nitrate. The protein fraction consists of enzymes such as amylase and lipase, as well as albumin, immunoglobulins and mucins, which also influence the viscosity of the saliva. As the amount of secretion increases, so does the secretion of bicarbonate and sodium ions, raising the pH of the saliva. In the literature, the pH of basal saliva is reported to be 5.8, while the pH of stimulated saliva is reported to be 7.6 [19,20].

The aim of this work was to further develop the existing model for biorelevant simulation of the esophagus [10,11]. One focus was to establish the temperature control of the model and the medium. In addition, the release behaviour of different film thicknesses and polymer qualities, in combination with highly and poorly soluble model drugs, was to be estimated and investigations with artificial saliva were to be carried out.

## 2. Materials and Methods

### 2.1. Materials

Various grades of polyvinyl alcohol, such as PVA 4-88, 8-88, 18-88 and 26-88 of the Emprove Essential series, were kindly provided by Merck (Darmstadt, Germany). Glycerol as plasticiser for the films was purchased from Caelo (Hilden, Germany) and demineralised water was used as solvent. Fluorescein sodium was purchased from Sigma-Aldrich Chemie (Darmstadt, Germany) and riboflavin from Fagron GmbH & CoKG (Barsbüttel, Germany). Potassium dihydrogen orthophosphate was obtained from neoFroxx GmbH (Einhausen, Germany) and sodium hydroxide was purchased from AppliChem GmbH (Darmstadt, Germany). For the preparation of artificial saliva, xanthan from VWR Chemicals (Radnor, PA, USA), mucin from Carl Roth GmbH & Co. KG (Karlsruhe, Germany), Tween 20 from Sigma-Aldrich Chemie GmbH (Steinheim, Germany), sodium hydrogen phosphate from neoLab Migge GmbH (Heidelberg, Germany) and anhydrous citric acid from Caesar & Loretz GmbH (Hilden, Germany) were used.

### 2.2. Methods

#### 2.2.1. Film Preparation

The films were prepared by the solvent casting technique [21]. For this purpose, 18.0 g of polyvinyl alcohol (4-88, 8-88, 18-88, 26-88), 2.0 g of glycerol (anhydrous), 80.0 g of purified water as solvent and 0.4 g of the model drugs were used to prepare the thin polymer films [10,11]. Polyvinyl alcohol 4-88, 8-88, 18-88 and 26-88 were used as polymers and fluorescein sodium and riboflavin were added as model drugs. All prepared formulations and further processing are shown in Table 1.

All components were mixed using a magnetic stirrer at 500 rpm in a laboratory glass bottle and then placed in a water bath at 80 °C for two hours with continuous stirring at 100 rpm. After the addition of the model drugs (fluorescein sodium, riboflavin), the mixture was stirred continuously at 80 °C for a further 60 min. For stability reasons, the following steps were carried out in the absence of light. The mixture was stirred overnight at 50 rpm and cooled to produce a bubble-free mixture.

The prepared polymer solution was then transferred to a motorised film casting device (mtv messtechnik CX4, Erftstadt, Germany). It was applied homogeneously to a polyethylene-coated liner using a doctors blade at different widths (500 µm, 1000 µm and 2000 µm). The films containing PVA 26-88 had to be coated at 50 °C due to the extremely high viscosity of the polymer solution. The films were dried overnight at room temperature (15–25 °C), then cut into elongated pieces with dimensions of 250 mm × 4 mm and stored in amber glass bottles at room temperature in the absence of light.

#### 2.2.2. General Structure of Biorelevant Release Simulation in EsoPeriDiss

The Esophagus Peristalsis Dissolution Tester, or EsoPeriDiss for short, was used for the biorelevant simulation of the esophagus [10,11]. The model, shown schematically in Figure 1, considers peristalsis, salivary flow rate, temperature control and possible patient positions.

The esophagus was simulated by a silicone tube with an internal diameter of 5 mm to represent the unfilled and collapsed state into which the polymer film under study was introduced. The release medium was introduced from a donor vessel into the simulated esophagus, similar to the open system of a flow cell, using a peristaltic pump that allowed the setting of biorelevant salivary flow rates of 0.5 mL/min in unstimulated mode and 6 mL/min in stimulated mode. As the esophagus is a very dry application site compared to the small intestine, fluid accumulation in the model was to be avoided, so a second peristaltic pump with a pump rate of 33 mL/min was used to transfer fluid from the model to the acceptor vessel. Two additional two-way taps were installed in the flow direction of the model to avoid negative pressure in the system and to allow other model media, such as syrup, to be introduced into the release model [11]. The model contains three of these silicone tubes in a parallel arrangement to allow triple determinations to be carried out simultaneously in all subsequent experiments.

A further development carried out as part of this work for the biorelevant characterisation of esophageal applied dosage forms was the complete temperature control of the model. The donor vessel was preheated on a laboratory hotplate or could be cooled with ice to simulate cold solutions. To address the lack of body temperature simulation in release experiments, the model was placed in an insulated and heated enclosure. A system of many simple fans providing high air exchange within the enclosure, combined with sensor controlled hot fans, maintained a uniform temperature of 37 ± 0.5 °C within the exposure model. Data loggers were used to record temperature profiles.

Esophageal peristalsis was simulated by a computer-controlled roller system that simulated peristalsis by squeezing the tubes. Six peristaltic events per hour were simulated in unstimulated mode and 180 peristaltic events per hour in stimulated mode. In addition, the model was simulated standing up in stimulated mode and lying down in unstimulated mode.

Samples were analysed spectrophotometrically using a fibre-optic system (Cary^®^60, Agilent Technologies, Santa Clara, CA, USA) with a 5 mm slit and base correction. Absorption in the acceptor vessel was measured every minute and the drug content calculated from the corresponding volumes; however, for better visualization, only every fifth value is shown below. Absorbance maxima and linearity of the absorbance range were determined. The analytical method was validated for linearity, precision, accuracy and selectivity. The non-eluted amount of drug relevant for local esophageal therapy was calculated and graphically displayed from the film weights measured prior to release and the absorptions from the release test [10,11]. The acceptor vessels were weighed empty before the test, the amount of medium added was also measured and the total weight of the acceptor vessels was measured after each dissolution test to allow accurate quantification of the drug content. A magnetic stirrer was used to ensure constant mixing during the dissolution tests. Due to the very good solubility of fluorescein sodium, only 200 mL of medium was added for technical reasons to allow fibre-optic measurement. As riboflavin has a very low solubility in water, 500 mL of medium was pre-filled into the acceptor vessel for all riboflavin dissolution tests so that the theoretically complete dissolution of the entire film in the pre-filled medium would result in a maximum concentration of approximately 10% with respect to the maximum solubility.

#### 2.2.3. Specific Release Test Parameters

In addition to the general release specifications, a variation of specific test parameters was adapted to assess relevant questions regarding the retention time of the mucoadhesive films. These additional parameters are shown in Table 2.

All studies on the influence of doctors blade widths of 500 µm, 1000 µm and 2000 µm on the release behaviour were carried out in unstimulated and stimulated modes with PVA 18-88 films loaded with fluorescein sodium. The influence of different polyvinyl alcohol grades (4-88, 8-88, 18-88, 18-88 + 26-88 (1:1), 26-88) on the release behaviour was investigated in unstimulated and stimulated mode with films loaded with the highly soluble drug fluorescein sodium and produced with a 1000 µm doctors blade. In addition, the influence of the release behaviour with the less soluble drug riboflavin in combination with the PVA grades (18-88, 26-88) produced with a 1000 µm doctors blade was investigated.

In addition, the lag time within the system was determined in each mode by adding 100 µL of concentrated fluorescein sodium solution into the three-way stopcock and measuring the time to absorbance detection using the fibre-optic system in the acceptor vessel [11].

#### 2.2.4. Temperature

Due to the temperature controllability of the release model, further investigations of the temperature-dependent release behaviour were carried out, but only in stimulated mode (Table 2), i.e., with a high flow rate based on liquid uptake. For this purpose, phosphate buffer pH 7.4 USP was tested ice-cooled, under physiological conditions at 37 °C and at 65 °C, which is the upper limit for pain-free tea drinking reported in the literature, based on usual and still possible drinking temperatures [22,23]. All investigations on the influence of temperature were carried out in stimulated mode with PVA 18-88 films loaded with fluorescein sodium and produced with a 1000 µm doctors blade.

#### 2.2.5. Artificial Saliva

The artificial saliva was developed based on the publication by Ali et al. [24]. It considers the pH, surface tension, viscosity and protein content of the artificial formulations and is adapted to unstimulated human saliva. Human saliva is a complex and variable medium. To identify the isolated influence of individual components on the retention time of esophageal applied mucoadhesive films, the developed buffer was tested step by step with increasing complexity so that the final formulation (SSF4) was analysed considering all simulated influencing factors. The basis was a British Pharmacopoeia citrate phosphate buffer (SSF1), shown in Table 3.

Tween^®^ 20 was added to modify the surface tension (SSF2) (Table 4). Xanthan was added in SSF3 to adjust the viscosity. Mucin was added to adjust the protein content (SSF4) and the xanthan was reduced to adjust the viscosity. The individual components were gradually added to the pure buffer solution SSF1. Xanthan gum and mucin were mixed with Tween^®^ 20 in a bowl using a pestle and gradually made up with buffer. All prepared SSFs were adjusted to pH 7.0 with 1 M HCl or NaOH. The solutions were freshly prepared for the experiments. All studies on the influence of artificial saliva were carried out in unstimulated mode with PVA 18-88 films loaded with fluorescein sodium and produced with a 1000 µm doctors blade.

## 3. Results and Discussion

### 3.1. Effect of Temperature on Release Performance

To improve the biorelevant simulation of the release behaviour of mucoadhesive films applied to the esophagus, the EsoPeriDiss release model was placed in an insulated housing and a sensor-controlled heating system was selected for constant temperature control in conjunction with an air circulation system.

The temperatures were recorded with temperature loggers during the heating phase and throughout the experiment. The heating phase, including the 60-min test, is shown symbolically in Figure 2A. The target temperature of 37 ± 0.5 °C was achieved 45 min and 60 min after the start of the heating phase in a stable plate condition. Other heating powers of up to 4000 W, also tested in advance, allowed in some cases shorter ramp-up phases at the beginning of the test but resulted in clearly noticeable variability in the plateau phase. For this reason, lower heating power was used for the further tests.

The possibility of temperature control in the model to physiological body temperatures led to an interest in investigating the residence times of the films under the influence of cold and hot media, and also to estimate the limits of the model. For this purpose, a series of experiments was carried out in which, in addition to the standard setup with phosphate buffer pre-tempered to 37 °C, cooled phosphate buffer and 65 °C hot buffer were used as the dissolution media at a high flow rate. In these cases, the high simulated flow rate of 6 mL/min should not be confused with the ingestion of cooled or hot drinks, but the results may provide trends in the retention time of the films with unauthorized drink ingestion after use of the EsoCap system, which should generally be taken at the edge of the bed. The amount of drug retained in the release model shown in Figure 2B shows a small but clearly visible trend outside the respective standard deviation. The shortest retention time was observed with the 65 °C buffer solution, followed by the 37 °C solution. The ice-cooled buffer solution followed last. After 14 min, only 10% of the drug dose was retained when simulating a high flow rate and using the 65 °C buffer. In each case, the release occurred 5 min later with the 37 °C buffer and then with the ice-cold solution.

As the polyvinyl alcohol used generally has a temperature-dependent dissolution rate and the viscosity of a polyvinyl alcohol–polymer solution also decreases with temperature, the release results also correspond to the substance-specific properties. However, no difference in the general lag time between the various test parameters at different temperatures could be determined. The relevance of the consistent further development of the model was clearly demonstrated with this test set-up. The enclosure and sensor-controlled temperature control of the model minimized the previous variability in the release experiments, e.g., due to fluctuating room temperatures, and thus further increased the robustness of the model results.

### 3.2. Artificial Saliva

A further improvement in the characterisation of biorelevant physiological release parameters after esophageal application of mucoadhesive films should be achieved by the development of artificial saliva as a release medium instead of the standard phosphate buffer USP pH 7.4 commonly used for intestinal release. There is already a large body of literature on artificial saliva, particularly about enzymatic activity and electrolyte composition [25,26]. However, as these parameters were not expected to affect the polyvinyl alcohol films used in previous release studies, the focus was on release-relevant parameters of physiologically unstimulated saliva [18,24,27]. The focus on unstimulated saliva is justified by the application of the EsoCap technology, which should preferably be applied at the edge of the bed before lying down. Unstimulated saliva, which is essentially the basal moisturizing secretion of the mouth and throat, is more viscous, has a lower pH and contains more mucins than saliva stimulated by smells or food, and was used as the reference for the development of the biorelevant medium [18]. Polysorbate 20 was added to reduce surface tension. Xanthan was used to increase the viscosity. Mucin was also used to influence viscosity but also to simulate the proteins in human saliva. Rather than develop a target formulation directly, a continuous formulation development was preferred, resulting in formulations SSF1–SSF4 and allowing the influence of each parameter on release behaviour to be estimated. The reduction in xanthan content between SSF3 and SSF4 (Table 4) was due to the viscosity, which would have been physiologically too high with a constant xanthan concentration and the addition of mucin. Therefore, during the development of SSF4, the viscosity was adjusted by reducing the xanthan content to achieve the target of 10 mg/mL mucin in the release medium. Salivary viscosity studies also showed shear thinning of the release medium, which is also found in human saliva [28]. However, a direct in vitro-in vivo correlation of the measured values and a comparison with the literature data for human saliva is difficult due to the variability of the literature values, which is also due to the difficulty in obtaining human saliva.

When the respective dissolution tests were performed (Figure 3), there was a grouping of results, both in the dissolution itself and in the determination of the lag times, and the two dissolution curves almost overlapped. The release tests of dissolution media SSF 1 and SSF 2 or SSF 3 and SSF 4 were almost congruent. The addition of the surfactant polysorbate to SSF2 showed no difference in release behaviour compared to the pure buffer solution of SSF1, but also no difference in lag time. However, the addition of xanthan gum to SSF3 resulted in a significant increase in retention time in the model, with the lag time also generally delayed by 4 min. The physiologically high viscosity of unstimulated saliva could therefore generally contribute to an increase in retention time of esophageal applied films. In contrast, modification by adding mucin to SSF4 showed no substantial difference in release and lag times compared to the already viscosity-adjusted release medium. However, the release studies with respect to the modification of the release medium confirm the general recommendation to apply the EsoCap system at the edge of the bed, followed by bed rest, which ideally leads to a physiological unstimulated saliva composition quickly and, as shown in previous studies, also brings the flow rate as close as possible to a standstill in order to allow a prolonged interaction of the active substance with the mucosa [11].

### 3.3. Effect of the Width of the Doctors Blade on the Retention Time

As mentioned above, the residence time of esophageal dosage forms is of great importance for therapeutic success, as the contact time of the drug with the mucosa appears to be directly correlated with therapeutic success [4,5]. Therefore, strategies to extend the retention time of esophageal dosage forms are of great interest. A simple way to increase the retention time could be to increase the film thickness by changing the width of the doctors blade. To characterise the influence of the doctors blade width during manufacture, the same fluorescein sodium-loaded polymer mixture of polyvinyl alcohol 18-88 was spread with doctors blade widths of 500 µm, 1000 µm and 2000 µm, and release tests were performed in unstimulated and stimulated mode using phosphate buffer pH 7.4 USP. The films produced with the 1000 µm doctors blade were 130 µm thick after drying, while the other films were half or twice as thick. In both unstimulated release mode (Figure 4A) and stimulated mode (Figure 4B), a substantial increase in film residence time was observed with increasing film thickness. In unstimulated release mode, only 10% of the films were retained in the model after 25 min with a blade width of 500 µm. An extension of 5 min was observed for films produced with a blade width of 1000 µm and a further extension of 24 min for films produced with a blade width of 2000 µm. In the stimulated release setup, only 10% of the films produced with a 500 µm blade width were still present after 10 min. After 19 min, 90% of the films produced with a 1000 µm blade width and 39% of the films produced with a 2000 µm blade width were released from the model.

As expected, the width of the blade during manufacture has a direct influence on the retention time of the polyvinyl 18-88 films tested in the model and, according to these results, thicker films are preferable for esophageal applications. However, it should be noted that these data cannot be directly transferred from the in vitro situation to the in vivo situation. On the one hand, any film used must first be applied to the esophagus, e.g., using application devices such as the EsoCap concept. However, there are also specific requirements for the application device [8,9]. For example, as the thickness of the film increases, so does the size of the roll of film placed in the EsoCap device and thus the cross-section of the capsule to be used. Patients with esophageal disease often report swallowing difficulties, which means that capsules with a large cross-section are not the preferred choice for these patients [29,30]. In addition, the mechanical properties of a thicker film are different and could pose a challenge during application. After the successful application of a thick film, the physiological self-cleaning reflex of the esophagus, which cannot be directly represented in the model, may need to be considered [15,31]. Interfering objects trigger additional peristaltic events in the esophagus with increased saliva production in the large salivary glands of the oral cavity, so a thick film that is designed to perform very well but potentially irritates the esophagus could compromise its own performance [32,33]. The adaptation of the unstimulated release mode to simulate secondary cleansing peristalsis is readily achievable, but the development of surrogate parameters to assess the irritation potential and thus elicit appropriate peristalsis is a challenge that should be addressed in the future to enable further biorelevant characterisation of esophageal dosage forms.

### 3.4. Effect of the Polymer Quality on Retention Time

As previously discussed, continuously increasing film thickness cannot be the key to success in prolonging the retention time of esophageal dosage forms. As an alternative, the modification of drug release by using different grades of polyvinyl alcohol was investigated. For this purpose, the same polymer solutions were always produced proportionally by the solvent casting method and spread with a doctors blade width of 1000 µm to form optically homogeneous film laminates. The PVA grades used in these tests all had a degree of hydrolysis of 88% and were therefore all soluble in aqueous media. The difference was mainly in the viscosity of the grades used due to the different molecular weights. These ranged from 32,000 g/mol for PVA 4-88 to 135,000 g/mol for PVA 26-88. Due to the different viscosities, the speed of the film drawing machine was adjusted during production.

The polymer blends of PVA 4-88 and PVA 8-88 were relatively low viscosity, so the take-off speed of the film stretching machine had to be increased from the standard speed of 10 mm/s. In addition, the viscosity of the PVA 26-88 polymer blend was so high that it could only be stretched at 50 °C. All formulations were released in phosphate buffer pH 7.4 USP in both unstimulated (Figure 5A) and stimulated (Figure 5B) modes.

As expected, the retention time of the films was longer in the unstimulated simulation than in the stimulated simulation. It is interesting to note that the results of the release tests in the different simulations showed very similar curves (Figure 5A,B). As expected, the greatest difference in the unstimulated mode was between PVA grade 4-88, which was 90% washed out of the simulated esophagus after 25 min, and PVA grade 26-88, which showed only 10% retention after 39 min. In the stimulated mode, the greatest difference was between PVA grades 4-88, which still had 10% retention after 7 min, and PVA 18-88, which was 90% eluted after 18 min.

However, a specific phenomenon was observed with PVA 26-88 films. These films could be removed completely and intact from the model after a successful release. Only under UV light did the films show a slight yellow colouration, due to minimal residual amounts of fluorescein sodium. Compared to the other polymers, which completely dissolved in the release medium within the test period, the polyvinyl 26-88 film was released by the diffusion of the highly soluble model drug fluorescein sodium. However, it can be assumed that both diffusion-induced and erosion-induced release effects occurred during the release tests, with the erosion-induced release probably having a significant influence on the release of the fast-swelling types such as polyvinyl alcohol 4-88 and, with increasing viscosity or chain length, the erosion-induced release component probably being the rate-determining component of the release. The modification of drug release, and thus the prolongation of the drug residence time on the esophageal mucosa, by varying the polymer quality is therefore not advisable, particularly in the case of highly water-soluble drugs.

Based on these studies, further investigations were carried out with riboflavin-loaded films. At 70 mg/L, riboflavin is virtually insoluble in water, unlike fluorescein sodium, which has a reported solubility of 500 g/L. The polyvinyl alcohol films produced in grades 18-88 and 26-88 showed an optically homogeneous appearance, although a very rough surface was observed compared to the films loaded with fluorescein sodium, which can be attributed to the suspended active ingredient.

The release profile (Figure 6) of the differently loaded polyvinyl alcohol 18-88 films shows a similar but clearly shifted release profile. The suspended and poorly soluble drug riboflavin was detected in the acceptor vessel significantly later than the highly soluble fluorescein sodium. While 40% of the fluorescein sodium was eluted from the simulated esophagus after 15 min, it took 10 min longer for 40% of the riboflavin to be released. This trend continued, so that after 28 min, only 10% of the drug remained in the esophagus, whereas after 42 min only 10% of the riboflavin remained in the esophagus.

The release profiles of the polyvinyl alcohol 26-88 films are again similar but show a clear shift. While the films loaded with sodium fluorescein showed a retention of only 10% after 37 min, the films loaded with riboflavin did not reach this value during the release tests. When the test was stopped after 90 min, 17% of the drug was still present in the film. In contrast to the fluorescein sodium films, the riboflavin films washed out of the esophageal silicone tubes after the release test all showed a clearly visible yellow colour.

A change in the release behaviour of the films applied to the esophagus is possible, as the data show. However, the solubility of the drug must be considered. There are several pharmaceutical technology options, especially for poorly soluble drugs such as those used in EoE, e.g., mometasone [34]. The possibility of using thin films and the associated ease of application, e.g., the EsoCap device, without changing the device cross-section, offer attractive adaptation mechanisms for controlling the specific release rate depending on the polymer. The question of whether thin films, which hardly or only very slowly erode at body temperature, trigger secondary peristalsis or whether the flexible structure of the films, as it was after release, makes them physiologically harmless, cannot be answered with the present model. As soon as the question of secondary peristalsis can be answered, biorelevant adjustments to the setup of the EsoPeriDiss can be made directly, as already explained.

## 4. Conclusions

The further development of the biorelevant release model with constant, sensor-controlled temperature regulation improves the reliability of the characterisation of new formulations for local, sustained drug delivery in the esophagus. Experiments with different temperatures and artificial saliva formulations showed that an increase in saliva viscosity, such as occurs at night, substantially prolongs the retention time, which is advantageous for the application of EsoCap films at the edge of the bed.

Strategies to increase the contact time of the dosage form with the esophagus were also investigated. Thicker films were found to be effective, especially for highly water-soluble substances, provided they could be combined with the application design. The influence of secondary peristalsis, triggered by a possible foreign body sensation in the esophagus, needs to be further investigated.

Varying the quality of the polyvinyl alcohol had little effect on the retention time for highly soluble substances but was a promising approach for prolonging the esophageal contact time for less soluble substances.

## Figures and Tables

**Figure 1 pharmaceutics-16-01021-f001:**
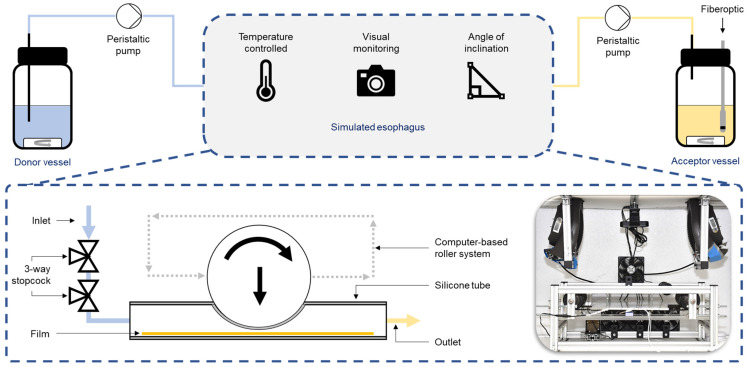
Schematic experimental setup of the biorelevant esophageal dissolution tester.

**Figure 2 pharmaceutics-16-01021-f002:**
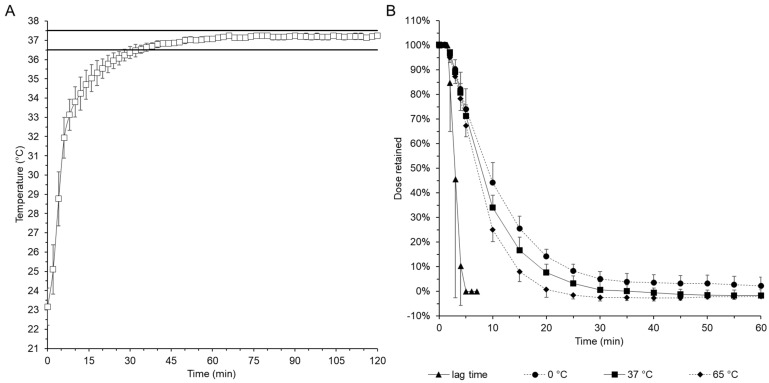
(**A**) Temperature profile recorded with a data logger inside the release system during the warm-up phase and the subsequent 60-min test phase. (**B**) The dose of fluorescein sodium retained during the investigation of the influence of different temperatures on the release medium in stimulated mode.

**Figure 3 pharmaceutics-16-01021-f003:**
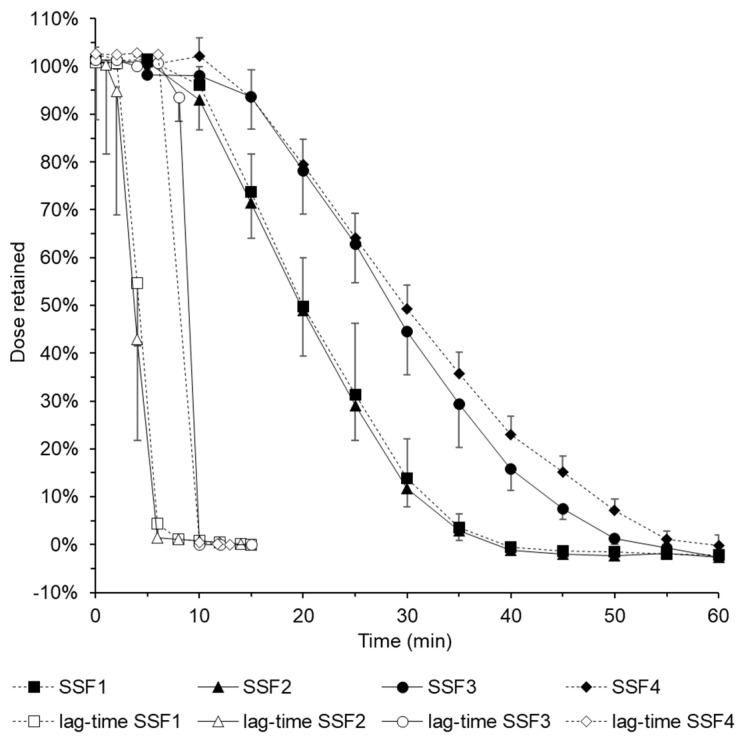
Dose retention of fluorescein sodium in the study of release behaviour with artificial saliva formulations.

**Figure 4 pharmaceutics-16-01021-f004:**
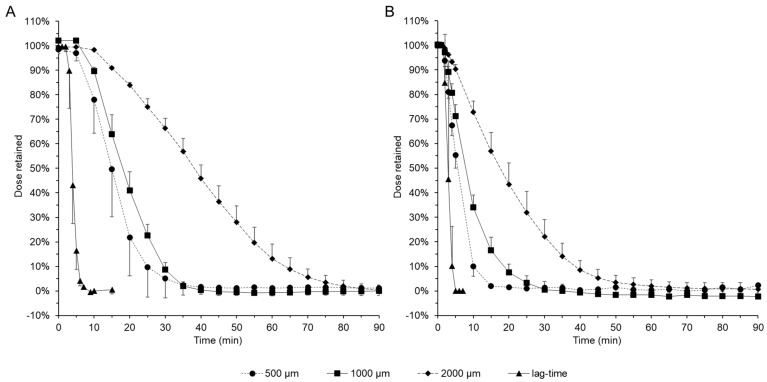
Influence of doctors blade width on the release behaviour of PVA films loaded with fluorescein sodium. (**A**) Unstimulated mode. (**B**) Stimulated mode.

**Figure 5 pharmaceutics-16-01021-f005:**
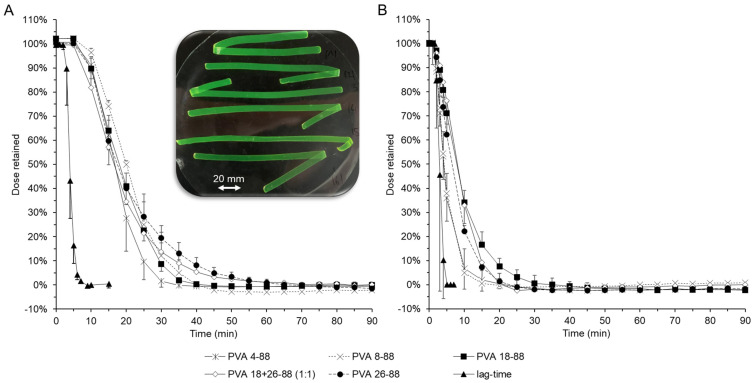
Influence of the type of polyvinyl alcohol used as base polymer on the release behaviour of films loaded with fluorescein sodium. (**A**) Unstimulated mode. Photo: Non-eroded films previously loaded with fluorescein sodium (PVA 18-88 + 26-88 and PVA 26-88) after the release test, slightly fluorescent under UV light. (**B**) Stimulated mode.

**Figure 6 pharmaceutics-16-01021-f006:**
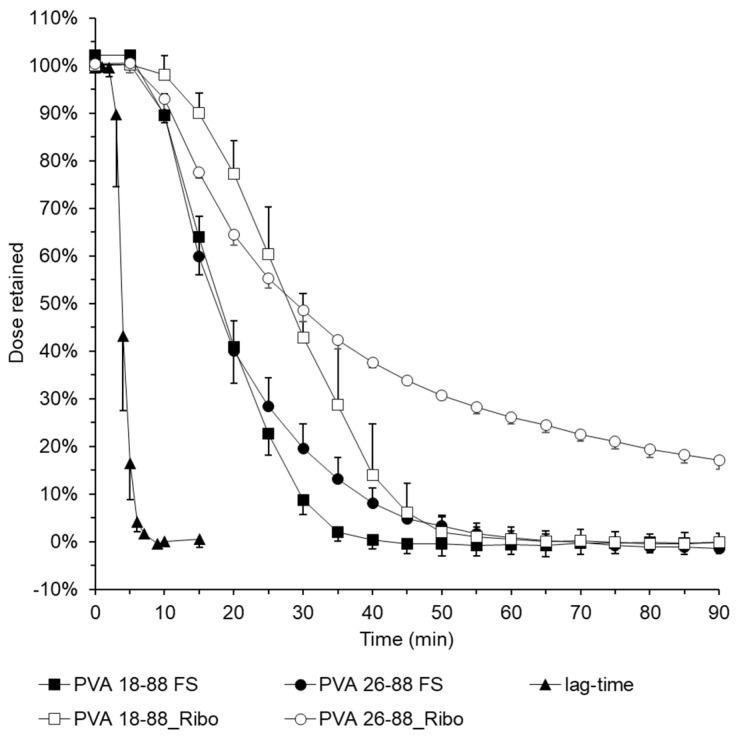
Influence of the solubility of the active ingredient used (FS: fluorescein sodium; Ribo: riboflavin) on the release behaviour of films using different types of polyvinyl alcohol as base polymer.

**Table 1 pharmaceutics-16-01021-t001:** Composition of the film formulations tested.

PVA Quality		4-88	8-88	18-88		26-88		18-88 + 26-88
polymer		18 g	18 g	18 g	18 g	18 g	18 g	9 g + 9 g
glycerol		2 g	2 g	2 g	2 g	2 g	2 g	2 g
water		80 g	80 g	80 g	80 g	80 g	80 g	80 g
API	fluorescein sodium	0.4 g	0.4 g	0.4 g	-	0.4 g	-	0.4 g
	riboflavin	-	-	-	0.4 g	-	0.4 g	-
doctors blade width		1000 µm	1000 µm	500 µm 1000 µm 2000 µm	1000 µm	1000 µm	1000 µm	1000 µm

**Table 2 pharmaceutics-16-01021-t002:** Overview of parameters and variations of formulations to extend mucosal contact time.

Mode	Unstimulated	Stimulated
angle	0° (lying down)	90° (upright)
flow rate	0.5 mL/min	6.0 mL/min
peristalsis	6/h	180/h
	3.7 cm/s	
temp. chamber	37.0 °C	
medium	phosphate buffer pH 7.4 USP	
	SSF 1-4	
temp. medium	All: 37 °C	
		PVA 18-88: +0 °C, +65 °C
PVA quality	4-88, 8-88, 18-88, 18-88 + 26-88 (1:1), 26-88	
doctors blade width	All: 1000 µm	
	PVA 18-88: +500 µm, +2000 µm	

**Table 3 pharmaceutics-16-01021-t003:** Composition of the citrate-phosphate buffer 7.0 BPC.

Substance	Quantity
citric acid, anhydrous	3.379 g
Na_2_HPO_4_ × 2 H_2_O	29.30 g
deionized water	add 1000.0 mL

**Table 4 pharmaceutics-16-01021-t004:** Overview of the composition for the step-by-step development of an artificial saliva.

Function	SSF1	SSF2	SSF3	SSF4
pH	7.692 mL buffer	7.692 mL buffer	7.692 mL buffer	6.211 mL buffer
surfactant		5.6 µL Tween^®^ 20	5.6 µL Tween^®^ 20	5.6 µL Tween^®^ 20
viscosity			0.08 *w*/*v*% xanthan	0.05 *w*/*v*% xanthan
viscosity and protein content				10 mg/mL mucin from pig stomach
medium	add 100.0 mL deionized water			

## Data Availability

The data supporting the results of this study are available upon request from the corresponding author, C.R., depending on the information requested.

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
