# Peer review of "Modification of the Biorelevant Release Testing of Esophageal Applied Mucoadhesive Films and Development of Formulation Strategies to Increase the Mucosal Contact Time"

_pharmaceutics, 2024, doi:10.3390/pharmaceutics16081021_

Round 1
Reviewer 1 Report
Comments and Suggestions for Authors
This study aimed to improve the existing model for biorelevant simulation of the esophagus and to assess the impact of polymer qualities and film thicknesses on the release of highly and poorly soluble model drugs. It is a very well-designed study, based on the previous results of the same team, but with the clear objective of further modifications of the model. The methods are contemporary and adequate for this type of study, and some new and significant results have been obtained, which brings novelty to the research field. There are only a few concerns that should be addressed, and they are as follows:
It needs to be clearer in ‘Methods’ section what already exists in the biorelevant dissolution model and what are the modifications of the model. Besides, since ‘Methods’ are before ‘Results’, some explanations are needed in ‘Methods’ section. For example, the abbreviation SSF is first time mentioned in Table 2. Later, in Table 4, it is not clear why there are 4 different saliva compositions (however, it was explained later in the text, line 231, but it should be clear in the Methods as well). Furthermore, it should be explained in Methods why there are two model substances, why there are 5 formulations with fluorescein sodium and 2 formulations with riboflavin for example.
Some figures should be clearer. For example, in Fig 2B it is difficult to see what refers to 0 and what to 65°C. It is the case also for Fig 4. I suggest changing the marker or the line options for one group.
The abstract should contain the main results and conclusion of the study.
Line 51: iodine or iodide?
There are also some technical issues:
Reference numbers should be before the full stop.
Some abbreviations are not defined in the text (e.g., GERD, EoE in line 22).
It should be ± instead of +/-.
Lag instead of LAG.
Comments on the Quality of English LanguageNo major issues were detected.
Author Response
Reviewer 1
We would like to thank the reviewer for his comments. The depth of understanding as well as the questions and discussions raised show a deep engagement with the topic and have helped us greatly to improve our work. As we have rarely received comments of such high quality and depth in the past, I would like to express my special thanks for your support of our scientific work. We have revised the manuscript considerably, especially in the methods section, but we have also been able to make many additions to other sections thanks to your comments.
It needs to be clearer in ‘Methods’ section what already exists in the biorelevant dissolution model and what are the modifications of the model.
- We have added a sentence to the methods section, for temperature control of the model.
Besides, since ‘Methods’ are before ‘Results’, some explanations are needed in ‘Methods’ section. For example, the abbreviation SSF is first time mentioned in Table 2. Later, in Table 4, it is not clear why there are 4 different saliva compositions (however, it was explained later in the text, line 231, but it should be clear in the Methods as well).
- We have restructured some sections and added other aspects to the methods section. In the section on saliva, we have added more information explaining why each component is present in artificial saliva. The structure and order of the chapters is determined by the journal.
Furthermore, it should be explained in Methods why there are two model substances, why there are 5 formulations with fluorescein sodium and 2 formulations with riboflavin for example.
- We have added explanations in the respective sections as to which films we analyzed in which configuration and hope that the supplementary texts together with the table provide a good overview of the release tests carried out.
Some figures should be clearer. For example, in Fig 2B it is difficult to see what refers to 0 and what to 65°C. It is the case also for Fig 4. I suggest changing the marker or the line options for one group.
- We have enlarged all the figures and enlarged the individual labels in the figures.
The abstract should contain the main results and conclusion of the study.
- We have added the results and more details on the study itself to the abstract
Line 51: iodine or iodide?
- Thanks.
There are also some technical issues:
Reference numbers should be before the full stop.
- Done
Some abbreviations are not defined in the text (e.g., GERD, EoE in line 22).
- Done
It should be ± instead of ±.
- Done
Lag instead of LAG.
- Done

Reviewer 2 Report
Comments and Suggestions for Authors
1. Suggested to reduce the similarity of the manuscript
2. Line. no. 64: Suggested to replace the word "amount" with "grades" of polyvinyl
3. Line no. 79: one grade of polyvinyl alcohol, suggested to specify the grade number in bracket.
4. Line no. 86: After the addition of the model drugs, suggested to indicate in bracket name of the model drug used.
5. Line no. 94: suggested to indicate the temperature for reproducibility as it vary depending different global location.
6. Does author performed the study in replicates, if yes then where is the stats statement?
7. Suggested to explain what is the need of citation in this statement "In this case, the high simulated flow rate of 6 mL/min should not be confused with the ingestion of cooled or hot drinks, but the results may provide trends in the retention time of the films with unauthorized drink ingestion after use of the EsoCap system.[22,23]".
8. Suggested to italicize the word "in vitro-in vivo" Line no. 238.
9. Suggested to increase the quality of the figure and reflect the error bar in both end i.e positive and negative
10. Suggested to provide the mechanical property data for the prepared all films tested.
11. Suggested to improve the discussion section, as at some instance I feel that authors tried to present mostly results, rather than interpreting them or correlating them with some either in vitro or in vivo results published previously.
12. Suggested to concise the conclusion, which is presently as summary of whole results.
Author Response
Reviewer 2
We would like to thank the reviewer for his comments. The depth of understanding as well as the questions and discussions raised show a deep engagement with the topic and have helped us greatly to improve our work. As we have rarely received comments of such high quality and depth in the past, I would like to express my special thanks for your support of our scientific work. We have revised the manuscript considerably, especially in the methods section, but we have also been able to make many additions to other sections thanks to your comments.
Line. no. 64: Suggested to replace the word "amount" with "grades" of polyvinyl
- Done
Line no. 79: one grade of polyvinyl alcohol, suggested to specify the grade number in bracket.
- Done
Line no. 86: After the addition of the model drugs, suggested to indicate in bracket name of the model drug used.
- Done
Line no. 94: suggested to indicate the temperature for reproducibility as it vary depending different global location.
- The European Pharmacopoeia defines the room temperature in the general chapters as 15 - 25 °C. We have added to this information. We have supplemented this information to match the usual room temperatures in our institute. For your information, the room temperature in our institute is usually between 20 - 22 °C, but we did not specifically monitor the room temperature during the production of the films.
Does author performed the study in replicates, if yes then where is the stats statement?
- In line 116, we had previously only explained that the model offers the possibility of a triple determination, without explicitly stating that we have of course used this possibility. We have added and clarified the sentence accordingly.
- The model contains three of these silicone tubes in a parallel arrangement, so that a triple determination could be carried out simultaneously in all subsequent experiments.
Suggested to explain what is the need of citation in this statement "In this case, the high simulated flow rate of 6 mL/min should not be confused with the ingestion of cooled or hot drinks, but the results may provide trends in the retention time of the films with unauthorized drink ingestion after use of the EsoCap system.[22,23]".
- You are right that the sources on the drinking temperature of liquids at this point contribute to a better understanding of the content elsewhere. Changes have been made accordingly.
Suggested to italicize the word "in vitro-in vivo" Line no. 238.
- Thank you for pointing this out. In the past, we have often written in vivo, in vitro in italics, but the journal Pharmaceutics has always changed this style, both in the review and in the submission process. Thank you for pointing this out, we have written in vivo, in vitro in italics throughout the manuscript and will pay particular attention to the style in the further process with the journal.
Suggested to increase the quality of the figure and reflect the error bar in both end i.e positive and negative
- We have enlarged all the figures and enlarged the individual labels in the figures. The standard deviations are still shown on one side only, to avoid overlapping the data as much as possible.
Suggested to provide the mechanical property data for the prepared all films tested.
- The focus of the work will be on the further development of the biorelevant release apparatus and the possibilities for extending the dwell time of the films applied in it. In addition, extensive work has already been carried out on the characterization of polyvinyl alcohol films:
- Jain, N.; Singh, V.K.; Chauhan, S. A Review on Mechanical and Water Absorption Properties of Polyvinyl Alcohol Based Composites/Films. Mech. Behav. Mater. 2017, 26, 213–222, doi:10.1515/jmbm-2017-0027.
Suggested to improve the discussion section, as at some instance I feel that authors tried to present mostly results, rather than interpreting them or correlating them with some either in vitro or in vivo results published previously.
- Before submitting this manuscript, we had a long internal discussion about the extent to which we should compare the discussion of the in vivo situation with the very short residence times (syrup less than 3 minutes). In the end, we decided not to do this and to focus on highlighting trends that could have an effect in vivo.
- From our point of view, it is important to highlight that an in vitro model should show trends, considering important influencing factors, but also has clear limitations. Because of the limitations, we have discussed very clearly, for example with the different film thicknesses, but also with the non-erosive PVA qualities, that a longer residence time is conceptually possible here, but other effects such as secondary peristalsis may have further influences. The occurrence of such peristalsis, for example, cannot be estimated with our model. For this reason, we believe that a direct in vivo in vitro correlation would be speculative, so we deliberately refrained from making a comparison and discussed the facts and possibilities available to us.
- With the trend results obtained, we will be able to carry out more targeted in vivo experiments, e.g. to evaluate the retention time in healthy volunteers.
Suggested to concise the conclusion, which is presently as summary of whole results.
- We have summarized the conclusions and hope that they now meet your expectations.

Reviewer 3 Report
Comments and Suggestions for Authors
The authors describe a further development of their biorelevant model for oesophageal films drug release and test several film formulations. The subject is interesting, well introduced and the results are well discussed.
Some specific points could be improved :
Please explicit acronyms when first cited (including GERD, EoE…).
Some sentences are rather long (discussion) : short conclusions are easier to follow for the reader.
Introduction :
Line 57 : please cite ref 10 and 11 after …existing model…
Methods section :
Please define LAG condition in methods section (API solution?)
How is the retained dose calculated (100%-released %, which is given by dosage)? Please define in method section.
How is riboflavin analyzed or the experimental setup adjusted? Spectrophotometry needs the drug to be dissolved in experimental conditions, which is not the case here, is it? Please specify in method section.
Which quality controls were performed on films before dissolution (eg mass uniformity, thickness measurements, humidity, textural properties…) to make sure there is no biais in dissolution results? Please include them in the article.
It remains often unclear which formulation was tested when and which results are presented i(n particular for Figures 2, 3). Table 1 is helpful, but naming your formulation in this table (eg formulation A, B, C…) and citing them in Figures legends and discussion paragraph associated would be easier for the reader.
Was the quantitative composition of formulations performed (eg 18% of PVA, 2% glycerol, 0,4% API…) – if based on previously formulated films, please cite the reference (ref 10 or 11?).
Results section :
When discussing all figures, it could be helpful to add a table with DT50 and DT90 (dissolution time 50% and dissolution time 90% => which corresponds to your 10% retained, commonly cited). One could expect that 10% release might not have a significant impact on clinical outcome, hence concluding on 10% only might be incomplete?
Please apply statistical tests to provide significance of results.
The DTx and statistica method shold also be introduced in method section.
Fig 4 : When the thickness of the film is increased, is the total quantity of formulation (API+excipients) the same or increased (please specify in method section)?
Why speaking about doctor blade height and not film thickness (final product specification)?
Fig 5 : please add the legend of the photo in panel A
What is the impact expected from diffusion-induced and erosion-induced formulations on local tolerance (compared to dissolving films)?
Fig 6 : what is the impact of low solubility of riboflavin on its delayed release – solubility problem or prolonged residence time? (cf question in methods section)
Comments on the Quality of English LanguageNone.
Author Response
Reviewer 3:
We would like to thank the reviewer for his comments. The depth of understanding as well as the questions and discussions raised show a deep engagement with the topic and have helped us greatly to improve our work. As we have rarely received comments of such high quality and depth in the past, I would like to express my special thanks for your support of our scientific work. We have revised the manuscript considerably, especially in the methods section, but we have also been able to make many additions to other sections thanks to your comments.
Some specific points could be improved :
Please explicit acronyms when first cited (including GERD, EoE…).
- Done
Some sentences are rather long (discussion) : short conclusions are easier to follow for the reader.
- We have shortened some particularly long sentences to make it easier to understand. Thank you for this very important feedback.
Introduction :
Line 57 : please cite ref 10 and 11 after …existing model…
- Done
Methods section :
Please define LAG condition in methods section (API solution?)
- The lag time was described in the general methods section. We have moved this section to the specific release test parameters section, explained it in more detail and added a source as the method is based on previous tests.
How is the retained dose calculated (100%-released %, which is given by dosage)? Please define in method section.
- We have added a corresponding paragraph in the method section on general release testing and referred to the earlier method development.
How is riboflavin analyzed or the experimental setup adjusted? Spectrophotometry needs the drug to be dissolved in experimental conditions, which is not the case here, is it? Please specify in method section.
- In addition, we have explained in the Methods section for which compound we added how much medium to the acceptor vessel and how we dealt with the poorer solubility of riboflavin to allow a UV-Vis analytical measurement, and that we ensured that in the theoretical 'worst case' a maximum riboflavin concentration of 10% relative to the maximum solubility was given.
Which quality controls were performed on films before dissolution (eg mass uniformity, thickness measurements, humidity, textural properties…) to make sure there is no biais in dissolution results? Please include them in the article.
- In the method section for determining the amount of drug released, we have added a section stating that we weighed each film prior to the release test.
It remains often unclear which formulation was tested when and which results are presented i(n particular for Figures 2, 3). Table 1 is helpful, but naming your formulation in this table (eg formulation A, B, C…) and citing them in Figures legends and discussion paragraph associated would be easier for the reader.
- Thank you for this comment, which was also made by other reviewers. For didactic reasons, we have decided not to use a central table structure similar to that used for the release data, and have added to the respective chapters in the Methods section which film materials were tested under which conditions, so that this structure can be found again in the Results and Discussion section, in the hope that the reader will gain a better understanding of the individual test setups without having to switch back and forth between the individual text sections.
Was the quantitative composition of formulations performed (eg 18% of PVA, 2% glycerol, 0,4% API…) – if based on previously formulated films, please cite the reference (ref 10 or 11?).
- Done
Results section :
When discussing all figures, it could be helpful to add a table with DT50 and DT90 (dissolution time 50% and dissolution time 90% => which corresponds to your 10% retained, commonly cited). One could expect that 10% release might not have a significant impact on clinical outcome, hence concluding on 10% only might be incomplete?
- Thank you for this suggestion. After analyzing further comments from reviewers, we have decided to increase the size of the figures to make them easier to read, and not to include a table with all the individual values. We feel that this central table would be of little use, especially as it would not include the release parameters, and that the existing figures, with their discussion of the influencing factors, are more useful to the reader.
- We have discussed the 10% limit many times in the past, in connection with previous publications and internally. As soon as we discuss the 0% value, we arrive at some impressively unrealistic retention times. Scintigraphy for the in vivo diagnosis of esophageal retention also shows clear analytical weaknesses in these low concentration ranges, so that the 0% value cannot be used to discuss the in vivo / in vitro correlation.
Please apply statistical tests to provide significance of results.
The DTx and statistica method shold also be introduced in method section.
- We believe that the statistical interpretation of early in vitro studies is of little use. We are very critical of interpreting low statistical significance from in vitro tests to the in vivo situation with its many confounding factors. However, we believe that early in vitro studies are an important tool for identifying initial trends that can be visually recognized when the results are appropriately presented. These trends can then be analyzed later, for example in studies on healthy volunteers, where of course we do static studies, especially because volunteers cause much greater variability in results.
Fig 4 : When the thickness of the film is increased, is the total quantity of formulation (API+excipients) the same or increased (please specify in method section)?
- We hope that the changes we have made to the methods section clarify our approach.
Why speaking about doctor blade height and not film thickness (final product specification)?
- We have measured the thickness of the films and supplemented the data in section 3.3 accordingly.
Fig 5 : please add the legend of the photo in panel A
- We have improved the picture caption.
What is the impact expected from diffusion-induced and erosion-induced formulations on local tolerance (compared to dissolving films)?
- We have clearly highlighted the limitations of our model in that we are unable to assess whether a non-eroding film or a very thick film could lead to secondary peristalsis and thus negatively affect retention time. Further investigation is required to identify relevant surrogate parameters. Once these are available, the relevant parameters in the model can be adjusted. At this stage, it can only be stated that depending on the solubility of the drug and the polymer used, it is potentially possible to increase the residence time of the drug. However, as the films are very flexible when wet and can be torn with relatively little force (not measured, only handling experience after release), we expect that the strong mechanical forces in the stomach will lead to destruction of the film structure and that there is therefore no risk of possible blockages in the gastrointestinal tract. However, these are only initial cautious estimates, which we cannot assess with absolute certainty, but which, as already explained, are based on our post-release handling experience with the relatively sensitive films.
Fig 6 : what is the impact of low solubility of riboflavin on its delayed release – solubility problem or prolonged residence time? (cf question in methods section)
- We have changed the aspect on solubility and corresponding quantification in the methods section.
- On your comment of solubility problem/prolonged residence time: After all the intensive discussions we have had with many clinicians over the last few years, I do not see the solubility of drugs embedded in the film matrix as a problem at all. From a clinical point of view, it is not a problem to get a drug into the esophagus at a high concentration for a short time; you can use dissolving tablets or syrups, but this high concentration is generally not very successful, as Hefner et al. have shown, for example. The challenge with esophageal therapy is time. The use of particularly potent drugs, such as glucocorticoids like mometasone or budesonide, which are even less soluble than riboflavin, results in a very positive clinical picture after local placement of an appropriate film containing the drug in the esophagus, as the data from the Aceso study show.

Round 2
Reviewer 2 Report
Comments and Suggestions for Authors
The authors have reflected all the said suggestions and comments, which made the manuscript enhanced with improved readability; Thus I suggest for further consideration with acceptance.